# From Genes to Disease: Reassessing *LOXHD1* and *AGBL1*’s Contribution to Fuchs’ Dystrophy

**DOI:** 10.3390/ijms26073343

**Published:** 2025-04-03

**Authors:** Tatiana Romanovna Tsedilina, Elena Ivanovna Sharova, Alexandra Vasilevna Kanygina, Boris Eduardovich Malyugin, Olga Pavlovna Antonova, Alexandra Vladimirovna Belodedova, Ivan Sergeevich Tkachenko, Aslan Mukhtarovich Gelyastanov, Andrey Vladimirovich Zolotarev, Aleksey Vladimirovich Klokov, Aleksandr Olegovich Murashev, Irina Viktorovna Fedyushkina, Edward Viktorovich Generozov, Liubov Olegovna Skorodumova

**Affiliations:** 1Medical Genomics Laboratory, Lopukhin Federal Research and Clinical Center of Physical-Chemical Medicine of the Federal Medical Biological Agency, 119435 Moscow, Russia; tsedilinat@gmail.com (T.R.T.); sharova78@gmail.com (E.I.S.);; 2The Jules Stein Eye Institute, University of California at Los Angeles, Los Angeles, CA 90095, USA; bmalyugin@mednet.ucla.edu; 3Department of Anterior Segment Transplant and Optical Reconstructive Surgery, S. Fyodorov Eye Microsurgery Complex Federal State Institution, 127486 Moscow, Russia; 4Samara Regional Clinical Ophthalmological Hospital named after T.I. Yeroshevsky, 443068 Samara, Russia; 5Krasnodar Branch, S. Fyodorov Eye Microsurgery Complex Federal State Institution, 350012 Krasnodar, Russia; 6Ophthalmology Clinical Centre “Bio Absolut”, 601909 Kovrov, Russia; 7Laboratory of Human Molecular Genetics, Lopukhin Federal Research and Clinical Center of Physical-Chemical Medicine of the Federal Medical Biological Agency, 119435 Moscow, Russiagenerozov@gmail.com (E.V.G.)

**Keywords:** Fuchs’ dystrophy, FECD, *AGBL1* variants, *LOXHD1* variants

## Abstract

Fuchs’ endothelial corneal dystrophy (FECD) is a genetically complex eye disease associated with multiple genes. A recent systematic review has raised concerns about the causal role of variants in the *LOXHD1* and *AGBL1* genes in the development of FECD. Conflicting data have been reported on the expression of the *LOXHD1* and *AGBL1* genes in the corneal endothelium. Furthermore, only partial segregation of the variants was observed in familial cases. An analysis of published datasets was conducted to examine the expression of *LOXHD1* and *AGBL1* genes in normal and FECD-affected corneal endothelia and progenitor cells. Neither *LOXHD1* nor *AGBL1* genes were expressed in normal or FECD corneal endothelia or progenitor cells. In-house cohorts were screened for carriers of previously reported *LOXHD1* and *AGBL1* variants. Carriers and their first-degree relatives were invited for an ophthalmological examination to reassess the causal relationship of these variants with FECD phenotype. Three carriers of *LOXHD1* variants (one carrier of rs200242497 and two carriers of rs192376005) and two carriers of *AGBL1* variants (rs181958589 and rs185919705) were recruited. None of the carriers or first-degree relatives over 50 years exhibited phenotypic signs of FECD via ophthalmic examination. The causal role of the *AGBL1* and *LOXHD1* variants found in the carriers was not confirmed. Taken together, our findings do not support a causal role for *AGBL1* and *LOXHD1* in the development of FECD.

## 1. Introduction

Fuchs’ endothelial corneal dystrophy (FECD) is a common degenerative eye disease characterized by the progressive loss of corneal endothelial cells and thickening of the Descemet’s membrane [1]. The disease develops slowly, usually over 20 years or more, leading to stromal edema and vision impairment [1]. The prevalence of FECD is approximately 7% of the population over 50 years of age [2].

FECD is classified into two clinical subtypes based on the age of onset. The early-onset form occurs within the first ten years of life [3], while the late-onset form affects patients over the age of 40 [4]. The early-onset form has an autosomal dominant inheritance pattern [3]. Late-onset FECD is also autosomal dominant but with variable penetrance and also occurs in sporadic forms [4,5]. In clinical practice, the most common form is the late-onset FECD, which is more prevalent in women [2,4]. No sex differences have been reported for early-onset FECD [6]. The etiology of FECD remains unclear, although several genes have been implicated. These genes include *TCF4*, *COL8A2*, *SLC4A11*, *ZEB1*, *AGBL1*, and *LOXHD1*, among others [7,8,9,10,11,12]. Below, we highlight important aspects of the causality of these genes in FECD.

The CTG18.1 trinucleotide repeat expansion (>40–50 repeats) in the intron region of the *TCF4* gene is thought to be causal for approximately two-thirds of late-onset FECD cases in the European ancestry population and has been confirmed in several independent cohorts, including the Russian cohort [8,13,14,15]. Nevertheless, variable penetrance and expressivity have been observed in FECD associated with the expansion of CTG18.1 repeats [13].

Mutations in the *COL8A2* gene have been implicated in early-onset FECD, including the missense variants NC_000001.11:g.36098318G>T (NP_005193.1:p.Gln455Lys), NC_000001.11:g.36098332A>C (NP_005193.1:p.Leu450Trp), and NC_000001.11:g.36098317_36098318delinsAC (NP_005193.1:p.Gln455Val), which segregate with the pathogenic phenotypes in familial cases [3,5,9].

Several studies have suggested an association between *SLC4A11* variants and FECD [16,17,18]. Three missense variants (NC_000020.11:g.3230954C>T (NP_114423.1:p.Glu399Lys), NC_000020.11:g.3228952C>T (NP_114423.1:p.Gly709Glu), NC_000020.11:g.3228687G>A (NP_114423.1:p.Thr754Met)) and one deletion NC_000020.11:g.3237581AG [1] (NP_114423.1:p.Pro34fs) were identified in a Chinese unrelated FECD cohort [16]. In a Northern European FECD cohort, seven missense variants were reported [17]. Of these, one variant, NC_000020.11:g.3228854C>T (NP_114423.1:p.Gly742Arg), was identified as segregating in a single family.

The *ZEB1* gene is actively expressed in the corneal endothelium and is associated with posterior polymorphous corneal dystrophy type 3 (PPCD3) [19,20,21]. Our systematic review of 14 studies identified *ZEB1* variants in FECD cases, but only one family showed partial segregation, and no variants were classified as pathogenic [10,22].

An association between variants in *AGBL1* and FECD has been proposed by Riazuddin et al. [11]. The NC_000015.10:g.86674435C>T (NP_689549.3:p.Arg1074*, hereinafter referred to as rs185919705) variant in the *AGBL1* gene was identified in a three-generation family with FECD. However, the segregation of the rs185919705 variant with the phenotype was only partial. All other variants described in the literature were found only in sporadic cases of FECD [22]. Data on the expression of *AGBL1* in the corneal endothelium are controversial. Riazuddin and co-authors demonstrated the presence of AGBL1 protein in the corneal endothelium of a patient with FECD via immunohistochemical (IHC) staining. However, several articles failed to find any RNA expression of the *AGBL1* gene [23,24,25].

Riazuddin et al. were the first to discover the association between *LOXHD1* variants and FECD in a familial case [12]. They performed linkage analysis using STR markers and identified a locus on chromosome 18 between the probes D18S484 and D18S1152 that was present in all affected members and absent in all unaffected members. Targeted sequencing revealed only one candidate variant, NC_000018.10:g.46591948G>A (NP_653213.6:p.Arg547Cys, hereinafter referred to as rs113444922), in the *LOXHD1* gene. However, this variant was localized outside of the locus defined via linkage analysis. Consequently, this variant was present in seven out of eight affected family members. Riazuddin et al. demonstrated LOXHD1 protein expression in the corneal endothelium via IHC staining [12]. The rs113444922 variant can now be considered as a non-segregating variant. In addition, other studies have reported no expression of the *LOXHD1* gene in the corneal endothelium [23,24,25].

In summary, the association of *LOXHD1* with FECD and the association of *AGBL1* with FECD are inconclusive. Variants in both genes did not segregate with a pathogenic phenotype in families with a hereditary history of FECD. The available data on gene products in the corneal endothelium are also controversial. Nevertheless, variants in *AGBL1* and *LOXHD1* are still mentioned in reviews as being associated with the disease [26,27,28,29]. Although these genes are mentioned in the review by Kannabiran et al., the authors suggest that the association of *LOXHD1* variants with FECD should be interpreted with caution [29]. In a systematic review, we have summarized all data on the *LOXHD1* and *AGBL1* genes in FECD and none of the variants were classified as pathogenic [22]. Therefore, further evidence is needed to determine the role of the *AGBL1* and *LOXHD1* genes in the etiology of FECD. This study focuses on investigating the involvement of the *AGBL1* and *LOXHD1* genes in the development of FECD.

## 2. Results

### 2.1. Experimental Design

It can be assumed that the pathogenicity of the *LOXHD1* and *AGBL1* gene variants results from the dysfunction or loss of protein function in the corneal endothelium. This disruption may occur during disease manifestation in adult tissue or during embryonic development. To address discrepancies in the assessment of *AGBL1* and *LOXHD1* gene expression, we used RNA-seq datasets and investigated their expression in the adult donor and FECD corneal endothelium samples from two independent datasets. Control group included 26 samples of donor corneal endothelium; FECD group included 24 samples of corneal endothelium from FECD patients (see Section 4 for details).

In addition, we analyzed the expression from several sets of progenitor cells, specifically human pluripotent stem cells (hPSCs), human embryonic stem cells (hESCs) (four samples from two independent sets), and human neural crest cells (hNCCs) samples (four samples from three independent sets), as corneal endothelium progenitors. hPSCs and hESCs are hereafter referred to as “stem cells”.

To investigate the correlation between FECD phenotype and the carriage of *AGBL1* and *LOXHD1* variants, we used the following strategy: First, we screened in-house cohorts for carriers of 20 rare, exonic, previously described FECD variants in *AGBL1* and *LOXHD1* (hereafter referred to as “candidate variants”; see Section 4 for details). We then performed ophthalmic examinations in carriers and their first-degree relatives over 50 years of age to assess the segregation of the variants with FECD.

### 2.2. No AGBL1 Expression in Progenitor Cells and Corneal Endothelium

The *AGBL1* gene showed trace-level expression in the control and FECD corneal endothelia from two independent datasets [30,31], as well as in its embryonic precursor cells (Figure 1). Conversely, the *GAPDH* gene exhibited a high level of expression in all transcriptome datasets, consistent with its housekeeping role. *TCF4* demonstrated a moderate-to-high level of expression, comparable to that of *GAPDH*, which is consistent with its role in neural crest cell migration. Thus, trace expression of *AGBL1* transcripts does not provide a sufficient explanation for the pathogenic effect of variants in *AGBL1* on the development of FECD through the loss of protein function.

### 2.3. Trace Expression of LOXHD1 in Corneal Endothelium and Its Progenitors

We examined the expression of the *LOXHD1* gene in a bulk RNA-seq dataset of both adult and progenitor corneal endothelium cells (hNCCs and stem cells). *LOXHD1* expression was not detected in either the FECD or control corneal endothelium from two independent datasets (Figure 2B) [30,31]. Assuming that the gene may play an important role in the development of the corneal endothelium, its expression level was examined in progenitor cells: hNCCs and stem cells. In contrast to *TCF4* and *GAPDH*, *LOXHD1* was not expressed in progenitor cells (Figure 1A and Figure 2A). In addition to *GAPDH* and *TCF4*, *SLC4A11* was chosen as a comparison gene because of its confirmed association with congenital hereditary endothelial dystrophy (CHED) accompanied by perceptive deafness [32]. *TCF4* and *SLC4A11* had target tissue expression levels comparable to *GAPDH* expression (Figure 1B and Figure 2B).

### 2.4. Candidate Variants Found Among In-House Cohorts

Our in-house previously genotyped cohorts were screened for carriers of 20 candidate variants in the *AGBL1* and *LOXHD1* genes (Table 1 and Appendix A). Five carriers were found and are referred to as “probands” (Table 1).

The presence of variants identified in probands from NGS or microarray genotyping data was confirmed via Sanger sequencing. To exclude the influence of the CTG18.1 trinucleotide repeat expansion on the development of FECD, the number of repeats in each proband was evaluated. None of the probands had an allele with more than 40 CTG18.1 trinucleotide repeats. These probands were then invited to participate in the present study (hereafter referred to as the “ophthalmic study”). All five probands signed an informed consent form and were enrolled in the ophthalmic study. All probands except RF17_172 underwent an ophthalmic examination. However, because this proband was younger than 50 years of age, we would have discounted the absence of FECD on ophthalmic examination anyway.

First-degree relatives over 50 years of age (mother, father, grandparents if available, and elder siblings) were invited to participate in the ophthalmic study. Relatives over 50 years of age who carried the candidate variant underwent an ophthalmic examination for signs of FECD. Relatives of four probands were enrolled in the ophthalmic study after signing an informed consent form.

In Table 2, we present the logic according to which we draw the conclusion from the results of the ophthalmic examination in the carriers.

### 2.5. No FECD Signs Among AGBL1 Candidate Variant Carriers

In the ophthalmic study group, there were two probands with variants in the *AGBL1* gene previously associated with FECD (Table 3). One proband (CTRL_001) was a carrier of the rs185919705 allele (NC_000015.10:g.86674435C>T, NP_689549.3:p.Arg1074*). He was also present in Cohort 2 as RF17-173. The mother of CTRL_001 (CTRL_001.2) also carried the rs185919705 allele. CTRL_001 and his mother were older than 50 years. The second proband (CTRL_048) was a carrier of rs181958589 (NC_000015.10:g.86674322G>C, NP_689549.3:p.Cys1036Ser) but younger than 50 years. She had a sister, also a carrier of rs181958589, who was 50 years old. Ophthalmic examinations of all carriers and relatives of the candidate *AGBL1* variants did not reveal any signs of FECD.

### 2.6. No FECD Signs Among LOXHD1 Candidate Variant Carriers

One proband (RF17-172) with the rs200242497 allele (NC_000018.10:g.46509805C>T, NP_653213.6:p.Glu1742Lys) was included in the ophthalmic study group (Table 3). He did not have an ophthalmic examination, but since he was younger than 50 years of age, the absence of FECD signs on ophthalmic examination would be inconclusive. However, his parents were included in the study. The father (RF17-172.3) was 58 years old and carried the rs200242497 allele. He had no evidence of FECD on ophthalmic examination.

Two probands carrying the rs192376005 allele (NC_000018.10:g.46592017G>A, NP_653213.6:p.Arg524Gly) were identified and included in the ophthalmic study group. One of them (RF17-23) had no older relatives, but he was 50 years old and did not have FECD. The second proband (RF17-63) was too young to draw conclusions from the ophthalmic findings. However, he did not have FECD based on the ophthalmic examination. His parents were also included in the study. His father (RF17-63.3) carried the rs192376005 allele. The father had a thorough ophthalmic examination, but no evidence of FECD was found. Thus, two cases confirm the absence of FECD in rs192376005 carriers.

## 3. Discussion

In this study, the causal role of variants in the *AGBL1* and *LOXHD1* genes in the development of FECD was investigated in two aspects. First, the expression levels of the genes of interest were analyzed in several available transcriptomic datasets. Second, carriers of variants in the *LOXHD1* and *AGBL1* genes without the *TCF4* gene repeat expansion were examined for the presence of the FECD phenotype.

The *AGBL1* gene encodes cytosolic carboxypeptidase 4, which is involved in the degradation of polyglutamate modifications in proteins. Polyglutamylation of tubulin is known to be maintained at high levels in microtubules. When the *AGBL1* gene was switched off, excessive polyglutamylation of tubulin was observed in Purkinje cells, which caused their neurodegeneration. Thus, it was concluded that controlling the length of polyglutamate residues is necessary to ensure the normal functioning of neurons [33]. However, the OMIM database did not indicate any other phenotypes caused by variants in *AGBL1* other than FECD [34].

The *LOXHD1* gene encodes the lipoxygenase homology domain 1 protein, which is conserved among vertebrates and consists of PLAT domains. Missense mutations in *Loxhd1* have been reported to cause hearing loss in mice. Mouse inner ear hair cells actively express both the *Loxhd1* gene and its protein [35,36]. In mouse embryos, *Loxhd1* gene expression was not detected in tissues other than the inner ear [35]. Subsequent analysis of the *Loxhd1* homolog in human families with autosomal recessive deafness led to the discovery of a nonsense variant that segregates with the disease. Indeed, many DFNB77 (deafness, autosomal recessive 77) families with complete segregation of *LOXHD1* variants have been reported [37]. *LOXHD1* has been shown to be important for mechanotransduction, but the mechanism is unknown [36]. Single-cell transcriptome analysis of E15 chicken embryo utricle cells has shown that the *Loxhd1* gene is specifically expressed in a small population of cells: striolar cells [38]. It is possible that single-cell transcriptome analysis of human utricle cells could also detect the expression of the *LOXHD1* gene in striolar cells and, thus, support the association of mutations in this gene with deafness.

A variant in a gene can cause the development of a disease if that gene is expressed in a mature target tissue or during the cell differentiation stage of that tissue. Data on the expression of *AGBL1* and *LOXHD1* in the corneal endothelium are conflicting in the literature. *AGBL1* expression was detected in a serial analysis of gene expression (SAGE) of donor and FECD-affected corneal endothelium samples [7]. Riazuddin et al. detected *LOXHD1* aggregates in the corneal endothelium and in the Descemet membrane of an FECD patient via IHC staining [12]. This suggested that the gene was expressed. Subsequent transcriptomic analysis of donor and FECD corneal endothelium samples in five studies failed to detect *AGBL1* expression [23,24,25,39,40]. Wieben et al. reported no expression of *AGBL1* and *LOXHD1* in FECD corneal endothelium samples (18 with the expansion of CTG18.1 repeats and 6 without the expansion of CTG18.1 repeats) [23]. In the paper by Chng et al., no expression was observed in the supplementary information on expression levels in ex vivo corneal endothelium samples from young and old donors and in cultured corneal endothelia [39]. Ali et al. reported no expression of *AGBL1* and *LOXHD1* in the results of transcriptome analysis of H9 hESC- and human PBMC-originated, iPSC-derived corneal endothelium cells [40]. Frausto et al. conducted microarray-based transcriptomic analysis of six pediatric and five adult donor corneas and reported no expression of *AGBL1* and *LOXHD1* [24]. In their later work, Frausto et al. confirmed the previous results: the *AGBL1* transcript was not detected via either RNA-seq or qPCR, while transcript levels for *LOXHD1* were only detected at trace levels [25].

Based on these results, Wieben et al. questioned the causality of variants in the *LOXHD1* and *AGBL1* genes [23]. Nevertheless, even the most recent reviews refer to variants in the *AGBL1* and *LOXHD1* genes as causal for FECD [41,42]. This may be due to the fact that none of the transcriptomic studies mentioned focused on elucidating the causality of variants in *AGBL1* and *LOXHD1*. This was not the main finding of these articles, and the question of the causality of variants in the *AGBL1* and *LOXHD1* genes remains unresolved.

In our work, we used the largest number of corneal endothelium transcriptomes (26 donor and 24 FECD affected) and explicitly showed that *AGBL1* and *LOXHD1* are expressed at trace levels. Thus, we confirmed the results of previous transcriptome analyses in independent sample sets. In addition, we tested the hypothesis that the expression of the *AGBL1* and *LOXHD1* genes in humans may be restricted to a specific stage of development. Therefore, we analyzed the gene expression of *AGBL1* and *LOXHD1* in different stem cell types and hNCCs, which are progenitors of corneal endothelial cells. No evidence of *AGBL1* or *LOXHD1* expression was detected in corneal endothelium progenitor cells.

It should be noted that the phenotypic manifestations of *SLC4A11* variants are similar to the declared manifestations of *LOXHD1* variants. Homozygous pathogenic *SLC4A11* variants are associated with CHED2 and Harboyan syndrome, and heterozygous pathogenic variants are associated with FECD [43,44]. At the same time, *SLC4A11* is confidently expressed in corneal endothelium, stem cells, and hNCCs (Figure 2A,B).

There are a few studies describing variants in the *AGBL1* and *LOXHD1* genes in FECD cases [11,12,15,45,46]. Most of these studies did not evaluate the expansion status of the CTG18.1 repeats in the *TCF4* gene. In this context, we attempted to re-evaluate the segregation of previously described rare exonic variants in the *LOXHD1* and *AGBL1* genes with the FECD phenotype. Our in-house cohorts were screened for the presence of previously described rare exonic variants in the *LOXHD1* and *AGBL1* genes. Carriers and their first-degree relatives over 50 years of age without CTG18.1 repeat expansion were examined for the signs of FECD.

Here, we discuss the limitations of our cohorts. For our work, it was most important to find carriers of the candidate variants in *AGBL1* and *LOXHD1* and the possibility to recall them and their relatives. For this purpose, we used our largest in-house cohorts. These cohorts are not representative of the general population. There is a bias toward athletes in our Cohort 2. Athletes have, among other traits, some genetic factors that allow them to perform exceptionally well in sports [47]. However, the association of variants in the *LOXHD1* and *AGBL1* genes with sports has not been reported. Furthermore, the presence of ophthalmic disease was not an exclusion criterion. The prevalence of FECD is known to vary by ethnicity, with the highest frequency observed in cohorts of European ancestry [48,49,50,51]. Almost all participants in our study cohorts were considered to be of European ancestry (see Section 4 for details). Thus, we could expect the presence of variants associated with late-onset ophthalmic diseases, including FECD.

Another limitation of our study is that the age cutoff for assessing the presence of FECD is 50 years. Results of ophthalmic examinations of participants (probands or relatives carrying candidate variants) older than 50 years were considered to draw conclusions (Table 2). The first signs of FECD (guttae) may appear later in life. However, in studies on the occurrence of cornea guttata, a significant increase in frequency was observed in the group above 40 years of age [52,53]. FECD prevalence studies have also used 50 years as a cutoff age to assess the presence of FECD [49,50,51].

Two rare exonic *AGBL1* variants that have been previously described in FECD were found in our in-house cohorts. The rs185919705 variant was re-evaluated for causality in one proband and one relative over 50 years of age. These participants did not have FECD. This variant has been described as causal in a familial case of FECD [11]. It was reported as an NM_152336.2:c.3082C>T substitution resulting in the formation of a premature stop codon. However, the segregation of the variant and the FECD phenotype in the family was only partial. This was discussed in detail in our systematic review [22]. The second *AGBL1* variant detected in our in-house cohort was rs181958589. It was detected in a participant over 50 years of age with no evidence of FECD. This variant was previously described in an FECD patient in the article by Riazuddin et al. [11] as NM_152336.2:c.2969G>C. It was also observed in one FECD patient (Dfu_90) in our cohort of FECD patients [15]. However, this FECD patient had an expanded CTG18.1 allele [15]. Thus, we have obtained evidence against the causality of all two variants described in the article by Riazuddin et al. [11], which initially suggested a role for *AGBL1* in the development of FECD.

Rare exonic variants in *LOXHD1*, previously reported in FECD, were screened in our in-house cohorts. The rs200242497 variant, which causes a glutamine-to-lysine substitution at amino acid residue 1742 of the NP_653213.6 *LOXHD1* isoform, was detected in a carrier over 50 years of age with no evidence of FECD. The second *LOXHD1* variant found in our cohort was rs192376005, which causes the NP_653213.6:p.Arg524Cys substitution. The absence of FECD in carriers of this variant was confirmed in two participants. These variants were first described in the article by Riazuddin et al. in FECD patients [12], each in one patient. Therefore, our results do not confirm a causal role of the rs200242497 and rs192376005 variants in FECD.

According to our systematic review, variants in *LOXHD1* and *AGBL1* (excluding benign and likely benign) were detected in 1% (4/400) and 0.7% (1/136) of FECD cases, respectively (in the combined FECD group) [22]. Similarly, variants in the *SLC4A11* and *ZEB1* genes (excluding benign and likely benign) were identified in 2.5% (16/675) and 0.7% (5/736) of FECD cases, respectively [22]. The frequency of FECD cases with CTG18.1 trinucleotide repeat expansion varied from 17% to 81.1% in different ethnicities [28]. Thus, in the broader context of FECD genetic research, excluding the contribution of variants in *LOXHD1* and *AGBL1* does not change the problem that there is no known cause for most FECD cases without CTG18.1 trinucleotide repeat expansion. In addition to specific genes associated with FECD, several chromosomal regions have been implicated. These include regions on chromosome 1 identified by Afshari et al., the 13pter-q12.13 region mapped by Sundin et al. in 2006, and the 5q33.1-q35.2 interval reported by Riazuddin et al. in 2009 [54,55,56]. However, no definitive causal relations between genes within these regions and FECD have been established, emphasizing the need for further research to elucidate the genetic basis of this condition.

In conclusion, we reported the expression levels of the *AGBL1* and *LOXHD1* genes at the zero level in the corneal endothelia of healthy donors, patients with FECD, and corneal endothelium progenitor cells. This did not confirm the possibility of involvement of *AGBL1* and *LOXHD1* gene products in the development of FECD, both by the mechanism of direct impairment of corneal endothelium function and by the mechanism of impairment of its formation during embryogenesis. No evidence of FECD was found in the carriers of variants suspected to be associated with FECD at the time of the ophthalmic examination. Considering the incomplete segregation observed in previously published familial cases and our results, it can be concluded that there is no evidence to support a role for the *AGBL1* and *LOXHD1* genes in the development of FECD.

## 4. Materials and Methods

### 4.1. Datasets for RNA-Seq Analysis

Transcriptome datasets were obtained for the following tissues:Adult human corneal endothelium samples from FECD patients and healthy donors (control samples) [30,31];hESCs and hPSCs [57,58];hNCCs [59,60].

A complete list of the datasets and samples analyzed is provided in the Appendix A.

### 4.2. RNA-Seq Analysis

RNA expression analysis included the following procedures: Initial assessment of sequencing quality was performed using FastQC (v0.11.9) [61] and multiQC (1.0.dev0) [62]. Cutadapt (3.7) was used to remove technical sequences, nucleotides, and reads of poor sequencing quality [63]. Sequencing quality was then re-assessed. The reads were mapped by Salmon (v1.8.0) to a complete transcriptome gencode.v43 using a transcriptome index with a full-genome decoy (GRCh38) [64]. RNA expression was then analyzed using the EdgeR package (3.28.1) [65] in RStudio (3.6.3) [66]. Data were normalized using the trimmed means of the M-values normalization method. The expression level for each gene was calculated in CPM (counts per million). To visualize the expression of the selected genes and transcripts, the ggplot2 package (3.3.6) was used [67].

### 4.3. List of Candidate Variants in the LOXHD1 and AGBL1 Genes

The list of *LOXHD1* and *AGBL1* variants, located in exons and splice sites with a MAF of less than 5% in gnomAD (v.2.1.1) for worldwide frequency or RuSeq [68,69], which was previously described in FECD, was taken from our systematic review [22]. These variants, referred to as “candidate variants”, can be viewed in Table 1 and Appendix A. This list included 3 unique variants in *AGBL1* and 17 unique variants in the *LOXHD1* gene. The accuracy of variant naming was verified using the web version of Variant Validator (Appendix A) [70].

### 4.4. Evaluation of the Correlation of the FECD Phenotype with the Carriage of the Candidate Variants

#### 4.4.1. Search Strategy for Carriers of *AGBL1* and *LOXHD1* Gene Variants

Our study has an observational design and was a cross-sectional study. We screened in-house cohorts for carriers of the 20 candidate variants in the *LOXHD1* and *AGBL1* genes (Table 1 and Appendix A). This search was conducted because we have the ability to recall patients for further study based on their informed consent. Carriers of candidate variants from our in-house cohorts were then invited to participate in the ophthalmic study. Candidate variants identified in probands were validated using Sanger sequencing.

Probands were also asked to invite their elder first-degree relatives to participate in the study. Relatives included in the ophthalmic study were genotyped for carriage of the proband’s variant. Probands and/or their relatives over 50 years of age with confirmed variant carrier status underwent ophthalmic examination for signs of FECD (details are provided below).

#### 4.4.2. Cohorts for Search for Variant Carriers

The present study sought to identify 20 candidate variants in two previously genotyped in-house cohorts. Cohort 1 (samples named CTRL_n) represented a part of a biomedical study, “Collection of control group samples for omics studies”, at the Lopukhin Federal Research and Clinical Center of Physical-Chemical Medicine of the Federal Medical Biological Agency (Lopukhin FRCC PCM FMBA of Russia) [71]. The objective of this study was to “obtain reference values of metabolites, microbiota composition, and DNA sequences in urine, blood, and feces samples of healthy volunteers for medical research in the field of genomics, metabolomics, proteomics, and other omics technologies”. Generally healthy, non-related individuals aged 18–75 of any sex were included in this cohort. Cohort 1 includes 50 participants, 28 of whom are female, with a mean age of 44.0 ± 12.6 years. The self-reported ethnicities were Russian (n = 45), Jewish (n = 2), Mordvin (n = 1), Gagauz (n = 1), and North Caucasian (n = 1) (Appendix A). Ancestry determined using Dodecad V3 resulted in a significant contribution of Western and Eastern European components (>70%) in all but one participant, who exhibited a non-European component as their primary ancestry, specifically Western Asia. The presence of eye diseases (other than acute inflammation) was not an exclusion criterion. Self-reported eye diseases included non-pathologic myopia (<6.00 diopters) (n = 13), pathologic myopia (>6.00 diopters) (n = 2), hyperopia (n = 3), astigmatism (n = 2), anterior uveitis in remission (n = 1), senile (age-related) cataract (n = 1), ectasia after keratorefractive surgery (n = 1), partial retinal detachment (n = 1), and vitreous body destruction (n = 1).

Cohort 2 participants were collected for the study by Semenova et al. (samples named RF17-n) [72]. Briefly, elite athletes, elite endurance athletes, and regularly training volunteers aged 18–65 of any sex were included in this cohort. Six participants from Cohort 2 were also in Cohort 1. Cohort 2 includes 171 participants, 56 of whom are female, with a mean age of 35.9 ± 7.9 years. The self-reported ethnicities were Russian (n = 153). The remaining participants were Ukrainian (n = 9), Belarusian (n = 3), Jewish (n = 2), Mordvin (n = 1), Mari (n = 1), Chuvashian (n = 1), and Tatar (n = 1). The participants were considered of European ancestry (Appendix A). The presence of eye disease was not an exclusion criterion. Information on the presence of eye disease was not collected.

### 4.5. Biological Samples and DNA Extraction

Blood or buccal swab samples were collected from participants after informed consent was obtained. DNA from blood was isolated using the Wizard Genomic DNA Purification Kit (Promega Corp.; Madison, WI, USA). Buccal swab DNA was extracted using the Gentra Puregene Blood Kit (Qiagen; Hilden, Germany) according to the manufacturer’s protocol.

### 4.6. Genotyping

DNA samples from Cohort 1 participants were processed for whole-exome sequencing. Exome libraries were prepared using the Ion Ampliseq Exome RDY-IC (ThermoFisher Scientific, Waltham, MA, USA) and Ion Xpress Barcode Adapters 1-16 Kits (ThermoFisher Scientific, Waltham, MA, USA) according to the manufacturer’s recommendations. Clonal amplification was performed using the Ion OneTouch2 System and Ion PI Hi-Q OT2 200 Kit consumables. Subsequently, sequencing was performed using the Ion Proton Sequencer (ThermoFisher Scientific, Waltham, MA, USA) with an IonPI v3 chip and Ion PI Hi-Q Sequencing 200 Kit (ThermoFisher Scientific, Waltham, MA, USA). The procedure of base calling and alignment was performed with the built-in sequencer tools (Torrent Suite) (ThermoFisher Scientific, Waltham, MA, USA) using the default analysis parameters. Variant calling was performed using cloud-based Ion Reporter v5.16 (Thermo Fisher, Waltham, MA, USA).

Cohort 2 participants were genotyped using Omni Express Exome 8 v1.4 (Illumina, San Diego, CA, USA). The raw data were processed using GenomeStudio software^®^ (2.0.4) (Illumina, San Diego, CA, USA), and a final report was obtained. Filtering steps, including removing alleles with a GC-score less than 0.3, were conducted. SNVs were deleted if they failed quality control in 10% and more samples. Only samples with less than 10% of failed SNVs quality were taken into analysis.

### 4.7. Validation of Variants and Analysis of TCF4 Repeat Expansion

CTG18.1 trinucleotide repeat expansion analysis was performed using a combination of short tandem repeat (STR) and triplet-primed PCR (TP-PCR) results. CTG18.1 trinucleotide repeats were scored as expanded if the number of repeats was >40. Primers for CTG18.1 repeat analysis, PCR conditions, and interpretation were described previously [15]. SNVs were genotyped via Sanger sequencing of the PCR products. Genotyping of rs192376005, rs181958589, and rs185919705 was performed using previously described primers (LOX-f/r1 and AGBL-f/r1) [15]. All primer sequences for SNP genotyping are shown in Appendix A. PCR, Sanger sequencing, and variant interpretation were conducted as described in Skorodumova et al. [15].

### 4.8. Ophthalmic Examination of Participants

A thorough anterior segment examination was performed on the available carriers and relatives who agreed to participate in the study (indicated in Table 1). Medical histories were obtained. Special attention was paid to the results of slit-lamp biomicroscopy (SL 120, Carl Zeiss Meditec AG, Jena, Germany). Specular microscopy (SP-2000P, Topcon, Tokyo, Japan) was also performed to calculate the endothelial cell density, shape, and size. The presence of corneal guttae was the main diagnostic criterion. FECD was also diagnosed even in the absence of clinical symptoms (such as decreased visual acuity, blurred vision in the morning hours, etc.). The stage of FECD was graded according to the modified Krachmer grading scale [4].

## Figures and Tables

**Figure 1 ijms-26-03343-f001:**
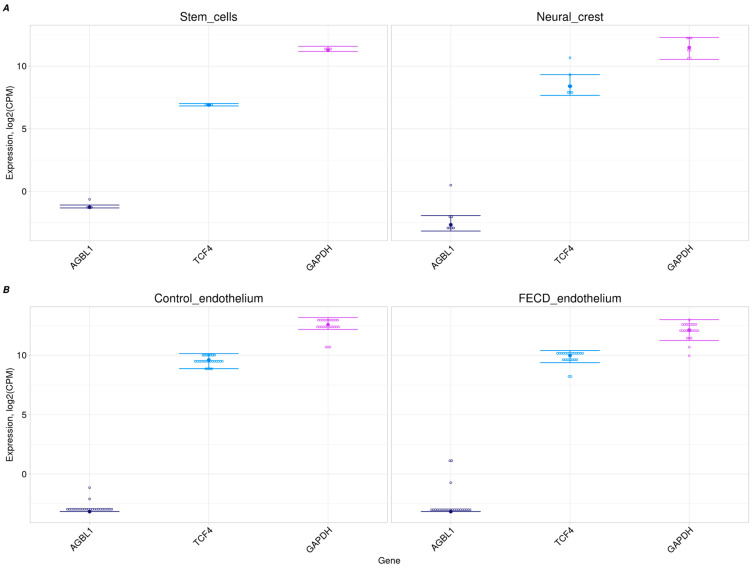
Expression of the *AGBL1* and comparison genes (*TCF4* gene, the neuronal cell marker, and the *GAPDH* gene). (**A**) In stem cells and hNCCs. (**B**) In control and FECD corneal endothelia.

**Figure 2 ijms-26-03343-f002:**
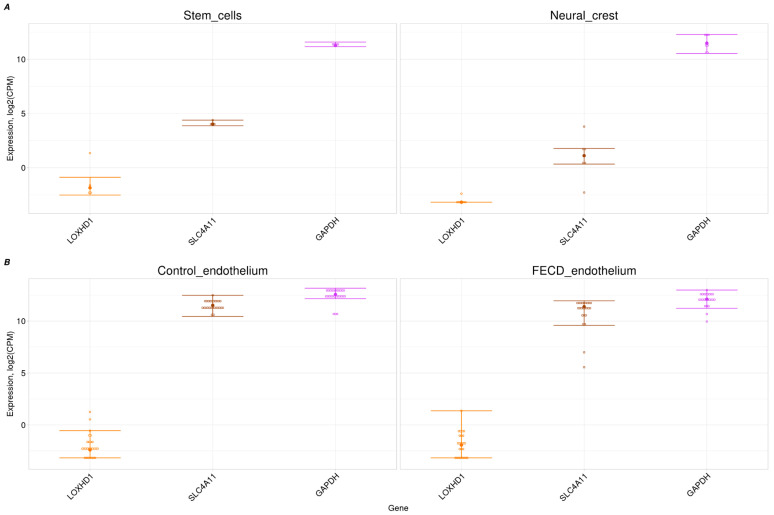
Bulk expression of *LOXHD1* and the comparison genes *GAPDH* and *SLC4A11.* (**A**) In progenitor cells (stem cells and hNCCs). (**B**) In control and FECD corneal endothelia.

**Table 1 ijms-26-03343-t001:** Allele frequency of candidate variants in Cohort 1, Cohort 2, and total gnomAD cohort. Abbreviations: n—number of participants in the cohort, ND—no data in the corresponding database.

Gene	dbSNP ID	HGVS Description of Candidate Variants	Allele Frequency in Cohort 1 (n = 50)	Allele Frequency in Cohort 2 (n = 171)	Allele Frequency in gnomAD (v2.1.1)
*LOXHD1*	rs148468627	NC_000018.10:g.46477695C>T	0 (0/100)	0 (0/342)	0.002227 (420/188598)
*LOXHD1*	rs764897088	NC_000018.10:g.46483630G>T	0 (0/100)	0 (0/342)	0.000006386 (1/156588)
*LOXHD1*	ND	NC_000018.10:g.46485062C>G	0 (0/100)	0 (0/342)	ND
*LOXHD1*	rs201994383	NC_000018.10:g.46507646G>A	0 (0/100)	0 (0/342)	0.0002981 (56/187842)
*LOXHD1*	rs775871086	NC_000018.10:g.46509757T>A	0 (0/100))	0 (0/342)	0.0002582 (49/189802)
*LOXHD1*	rs200242497	NC_000018.10:g.46509805C>T	0 (0/100)	0.00292 (1/342)	0.0007381 (140/189686)
*LOXHD1*	rs372241056	NC_000018.10:g.46521131A>G	0 (0/100)	0 (0/342)	0.0004314 (81/187742)
*LOXHD1*	rs200792636	NC_000018.10:g.46541815G>A	0 (0/100)	0 (0/342)	0.0002001 (36/189872)
*LOXHD1*	rs564297037	NC_000018.10:g.46566335G>A	0 (0/100)	0 (0/342)	0.00005664 (9/158890)
*LOXHD1*	rs376539851	NC_000018.10:g.46566443G>A	0 (0/100)	0 (0/342)	0.0005068 (95/187466)
*LOXHD1*	rs141932807	NC_000018.10:g.46577732C>T	0 (0/100)	0 (0/342)	0.001258 (238/189162)
*LOXHD1*	rs540100675	NC_000018.10:g.46579680G>A	0 (0/100)	0 (0/342)	0.00009475 (18/189970)
*LOXHD1*	rs113444922	NC_000018.10:g.46591948G>A	0 (0/100)	0 (0/342)	0.00007337 (14/190808)
*LOXHD1*	rs192376005	NC_000018.10:g.46592017G>A	0 (0/100)	0.00585 (2/342)	0.002651 (509/192004)
*LOXHD1*	rs566553343	NC_000018.10:g.46639658G>A	0 (0/100)	0 (0/342)	0.00003192 (6/187960)
*LOXHD1*	rs2039074525	NC_000018.10:g.46649158C>T	0 (0/100)	0 (0/342)	ND
*LOXHD1*	rs980201296	NC_000018.10:g.46649241A>C	0 (0/100)	0 (0/342)	0.00001269 (2/157626)
*AGBL1*	rs185919705	NC_000015.10:g.86674435C>T	0.01000 (1/100)	0.00292 (1/342)	0.001753 (486/277246)
*AGBL1*	rs377248005	NC_000015.10:g.86397372G>A	0 (0/100)	0 (0/342)	0.0002209 (60/271636)
*AGBL1*	rs181958589	NC_000015.10:g.86674322G>C	0.01000 (1/100)	0 (0/342)	0.001130 (312/276112)

**Table 2 ijms-26-03343-t002:** Logic of conclusions from the results of ophthalmic examination in carriers.

Participant	Candidate Variant Carriage	Age, Years	Ophthalmic Examination Results	Conclusion
proband	carrier	>50	no FECD	causality rejected
		>50	FECD	causality not rejected
		<50	no FECD	inconclusive
		<50	FECD	causality not rejected
first-degree relative	carrier	>50	no FECD	causality rejected
		>50	FECD	causality not rejected
	non carrier	>50	no FECD	inconclusive
		>50	FECD	inconclusive

**Table 3 ijms-26-03343-t003:** Information about carriers found in Cohorts 1 and 2. Abbreviations: ND—no data as examination not performed.

Participant ID	Relation	Proband ID	Sex	Age	*AGBL1*		*LOXHD1*		CTG18.1 Genotype (Repeats Number in Each Allele)	Ophthalmic Examination	FECD Diagnosis (Krachmer Grade)
					rs185919705	rs181958589	rs192376005	rs200242497			
CTRL_001	proband	CTRL_001/RF17-173	male	51	T/C (het)				ref hom (14/26)	yes	no/(0)
CTRL_001.2	mother	CTRL_001	female	80	T/C (het)				ref hom (11/14)	yes	no/(0)
CTRL_001.3	father	CTRL_001	male	78	C/C (ref hom)				ref hom (11/26)	yes	no/(0)
CTRL_048	proband	CTRL_048	female	48		C/G (het)			ref hom (11/11)	yes	no/(0)
CTRL_048.4	sister	CTRL_048	female	50		C/G (het)			ref hom (11/31)	yes	no/(0)
RF17-023	proband	RF17_023	male	50			A/G (het)		ref hom (22/27)	yes	no/(0)
RF17-063	proband	RF17_063	male	37			A/G (het)		ref hom (11/14)	yes	no/(0)
RF17-063.2	mother	RF17-063	female	59			A/A (ref hom)		ref hom (14/17)	yes	no/(0)
RF17-063.3	father	RF17-063	male	61			A/G (het)		ref hom (11/34)	yes	no/(0)
RF17-172	proband	RF17_172	male	30				T/C (het)	ref hom (14/15)	no	ND
RF17-172.2	mother	RF17_172	female	53				C/C (ref hom)	ref hom (11/15)	yes	no/(0)
RF17-172.3	father	RF17_172	male	58				T/C (het)	ref hom (14/24)	yes	no/(0)

## Data Availability

Gene expression datasets were obtained from the open databases NCBI SRA (https://www.ncbi.nlm.nih.gov/sra/, accessed on 10 January 2022) and Gene Expression Omnibus (https://www.ncbi.nlm.nih.gov/gds/, accessed on 10 January 2022). Accession numbers are provided in Appendix A. Raw whole-exome data from participants in Cohort 1, microarray data from participants in Cohort 2 are not available. The original contributions presented in the study are included within the article/Appendix A, and further inquiries can be directed to the corresponding author.

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
