# Peer review of "From Genes to Disease: Reassessing LOXHD1 and AGBL1’s Contribution to Fuchs’ Dystrophy"

_ijms, 2025, doi:10.3390/ijms26073343_

Round 1
Reviewer 1 Report
Comments and Suggestions for Authors
This is very well written and conducted study that supports exclusion of variants in LOXHD1 and AGBL1 as a cause of Fuchs endothelial corneal dystophy.
Authors used several different aproaches including RNA-seq analysis, systematic review of up-to-date reported variants in these two genes, genotype-phenotype correlations, etc. to support their conclusions.
I would prefer description of variants on "DNA" level instead of protein but is it only a minor comment.
Author Response
We thank the reviewer for appreciation of the importance of this study and the valuable comment. We have added "DNA" level descriptions for each mentioned variant in the main manuscript file. They are indicated in yellow.
Reviewer 2 Report
Comments and Suggestions for Authors
- While the manuscript mentions that the cohorts are not representative of the general population, a clearer justification for this limitation would help readers better understand the potential implications of the findings for broader populations.
- While the absence of FECD signs in variant carriers is compelling, it would be beneficial to discuss more thoroughly the potential reasons for phenotypic variability.
- It would benefit from a more in-depth discussion of how the present study's results fit within the broader context of FECD genetic research, particularly concerning the role of other known genes associated with the disease.
- Additionally, explaining how the cohort sizes were determined would be useful to assess the power of the study.
- It could also be helpful to discuss the implications of these findings for future genetic research in FECD and other similar corneal dystrophies.
Author Response
We greatly appreciate your comments and suggestions to improve our manuscript. Please find our response on your queries here below.
Additional or corrected text in the main manuscript file indicated in yellow.
Comments 1: While the manuscript mentions that the cohorts are not representative of the general population, a clearer justification for this limitation would help readers better understand the potential implications of the findings for broader populations.
Response 1: It was noted that further elucidation is required to ensure comprehensive understanding of the research's limitations, particularly the fact that the study groups do not represent the general population. We would like to thank the Reviewer for pointing this out. This issue is addressed in lines 295 through 305 of the revised manuscript. The text highlights the bias towards athletes in Cohort 2 and the predominantly European ancestry of the majority of the participants:
"There's a bias toward athletes in our Cohort 2. Athletes have, among other traits, some genetic factors that allow them to perform exceptionally well in sports [47]. However, the association of variants in the LOXHD1 and AGBL1 genes with sports has not been reported. Furthermore, the presence of ophthalmic disease was not an exclusion criterion. The prevalence of FECD is known to vary by ethnicity, with the highest frequency observed in cohorts of European ancestry [48-51]. Almost all participants in our study Cohort 1 are of European ancestry, as evidenced by the significant contribution of Western and Eastern European Dodecad components. Similarly, almost all self-reported ethnicities of Cohort 2 participants were also European. Thus, we could expect the presence of variants associated with late-onset ophthalmic diseases, including FECD."
Comments 2: While the absence of FECD signs in variant carriers is compelling, it would be beneficial to discuss more thoroughly the potential reasons for phenotypic variability.
Response 2: We are grateful for this valuable comment. In order to discuss possible causes of phenotypic variability in more detail, we have incorporated additional information regarding the variability of FECD among various ethnic groups. The highest observed frequency of FECD was identified in cohorts of European ancestry. The composition of our study cohorts is predominantly European, reflecting the demographic characteristics of the studied populations. This information has been included in lines 299-305 of the Discussion section of the revised manuscript:
"The prevalence of FECD is known to vary by ethnicity, with the highest frequency observed in cohorts of European ancestry [48-51]. Almost all participants in our study Cohort 1 are of European ancestry, as evidenced by the significant contribution of Western and Eastern European Dodecad components. Similarly, almost all self-reported ethnicities of Cohort 2 participants were also European. Thus, we could expect the presence of variants associated with late-onset ophthalmic diseases, including FECD. "
Comments 3: It would benefit from a more in-depth discussion of how the present study's results fit within the broader context of FECD genetic research, particularly concerning the role of other known genes associated with the disease.
Response 3: We agree that this discussion will improve our manuscript. The present study's results have been integrated with the broader context of FECD genetic research, as detailed in lines 337-351 of the Discussion section of the revised manuscript:
"According to our systematic review, variants in LOXHD1 and AGBL1 (excluding benign and likely benign) were detected in 1% (4/400) and 0.7% (1/136) of FECD cases, respectively (in the combined FECD group). Similarly, variants in SLC4A11 and ZEB1 genes (excluding benign and likely benign) were identified in 2.5% (16/675) and 0.7% (5/736) of FECD cases, respectively. The frequency of FECD cases with CTG18.1 trinucleotide repeat expansion varied from 17% to 81.1% in different ethnicities [28]. Thus, in the broader context of FECD genetic research, excluding the contribution of variants in LOXHD1 and AGBL1 does not change the problem that there is no known cause for most FECD cases without CTG18.1 trinucleotide repeat expansion. In addition to specific genes associated with FECD, several chromosomal regions have been implicated. These include regions on chromosome 1 identified by Afshari et al., the 13pter-q12.13 region mapped by Sundin et al. in 2006, and the 5q33.1-q35.2 interval reported by Riazuddin et al. in 2009. However, no definitive causal relations between genes within these regions and FECD have been established, emphasising the need for further research to elucidate the genetic basis of this condition".
Comments 4: Additionally, explaining how the cohort sizes were determined would be useful to assess the power of the study.
Response 4: While we agree that this clarification would have been beneficial, we were unable to implement it due to the design of our study and the characteristics of the variants investigated. We have explained the reasons in more detail below. The purpose of our study was to use in-house cohorts to identify individuals carrying genetic variants of interest. We used the maximum sample size available to us, meeting the requirement that the frequency of carriers is not reduced relative to the population frequency due to known factors, and that it is possible to invite relatives over 50 years of age. This was mentioned in the Discussion section in lines 295-305.
The study design included only one group. There weren't two groups to compare - a group with FECD and a control group. The study did not aim to make comparisons between groups, such as a group with FECD and a control group. The objective of our study was not to draw conclusions about allele frequencies of variants.
All variants described in the study in AGBL1 and LOXHD1 genes are rare. While the cutoff was set at 5%, it's important to note that the variants included have a frequency of 0.27% or less in GnomAD (https://gnomad.broadinstitute.org/). We will also present the calculation of the minimum sample size for both groups (allocation 1:1) to reach 80% power. For example, in the FECD group, we could detect variants with an allele frequency of 0.5% (1/200), similar to our previous study (https://doi.org/10.1167/iovs.18-24590). The allele frequency in the control group could be set at 0.27%, which is the maximum among variants of interest. In this case, to reach 80% of power, the sample size for both cohorts would require 1,500 participants. For rarer variants, the sample size would need to be substantially larger, potentially reaching tens of thousands of participants. Achieving meaningful power is only feasible within the scope of extensive genetic projects. For these reasons, our study did not have a case-control design.
Comments 5: It could also be helpful to discuss the implications of these findings for future genetic research in FECD and other similar corneal dystrophies.
Response 5: We agree that this discussion will improve our manuscript. We have added the discussion on the implications of these findings for future genetic research in FECD and other similar corneal dystrophies in lines 345-359 of the Discussion section of the revised manuscript:
"The variants considered in our study are an example of a case where the variants are rare but not pathogenic or causal. This problem is relevant not only for the study of hereditary corneal dystrophies, but also for rare hereditary diseases in general. One of the main filters used in the search for candidate variants is population frequency. However, it is important to note that rarity is not sufficient to give a variant pathogenic status; it is only one of the criteria [57]. Further experiments are required to provide evidence for its pathogenicity status. Exclusion of known factors in the development of the disease under study in the probands, verification of gene expression in the target tissues, and adherence to medical-genetic guidelines for variant interpretation will help minimize the establishment of false associations between variant and disease."
Round 2
Reviewer 2 Report
Comments and Suggestions for Authors
The authors have responded to my concerns
Author Response
We are very grateful for your suggestions. They have been very helpful indeed in improving our manuscript.